# Domains and Functions of Spike Protein in SARS-Cov-2 in the Context of Vaccine Design

**DOI:** 10.3390/v13010109

**Published:** 2021-01-14

**Authors:** Xuhua Xia

**Affiliations:** 1Department of Biology, University of Ottawa, Marie-Curie Private, Ottawa, ON K1N 9A7, Canada; xxia@uottawa.ca; Tel.: +1-613-562-5718; 2Ottawa Institute of Systems Biology, University of Ottawa, Ottawa, ON K1H 8M5, Canada

**Keywords:** COVID-19, spike protein, S-2P, SARS-CoV-2, cleavage, vaccine, protein structure, hydrophobicity, isoelectric point

## Abstract

The spike protein in SARS-CoV-2 (SARS-2-S) interacts with the human ACE2 receptor to gain entry into a cell to initiate infection. Both Pfizer/BioNTech’s BNT162b2 and Moderna’s mRNA-1273 vaccine candidates are based on stabilized mRNA encoding prefusion SARS-2-S that can be produced after the mRNA is delivered into the human cell and translated. SARS-2-S is cleaved into S1 and S2 subunits, with S1 serving the function of receptor-binding and S2 serving the function of membrane fusion. Here, I dissect in detail the various domains of SARS-2-S and their functions discovered through a variety of different experimental and theoretical approaches to build a foundation for a comprehensive mechanistic understanding of how SARS-2-S works to achieve its function of mediating cell entry and subsequent cell-to-cell transmission. The integration of structure and function of SARS-2-S in this review should enhance our understanding of the dynamic processes involving receptor binding, multiple cleavage events, membrane fusion, viral entry, as well as the emergence of new viral variants. I highlighted the relevance of structural domains and dynamics to vaccine development, and discussed reasons for the spike protein to be frequently featured in the conspiracy theory claiming that SARS-CoV-2 is artificially created.

## 1. Introduction

SARS-CoV-2 uses its trimeric spike protein for binding to host angiotensin-converting enzyme 2 (ACE2) and for fusing with cell membrane to gain cell entry [1,2,3,4]. This is a multi-step process involving three separate S protein cleavage events to prime the SARS-2-S for interaction with ACE2 [2,3], and subsequent membrane fusion and cell entry. These processes involve different domains of the S protein interacting with host cell and other intracellular and extracellular components. Efficiency in each step could contribute to virulence and infectivity. Disrupting any of these steps could lead to medical cure.

The domain structure is very similar between SARS-S (UniProtKB: P59594) and SARS-2-S (UniprotKB: P0DTC2). Both are cleaved to generate S1 and S2 subunits at specific cleavage sites (Figure 1A). S1 serves the function of receptor-binding and contains a signal peptide (SP) at the N terminus, an N-terminal domain (NTD), and receptor-binding domain (RBD). S2 (Figure 1A) functions in membrane fusion to facilitate cell entry, and it contains a fusion peptide (FP) domain, internal fusion peptide (IFP), two heptad-repeat domains (HR1 and HR2), transmembrane domain, and a C-terminal domain [2,3,5,6,7,8]. However, there are also significant differences between SARS-S and SARS-2-S. For example, the contact amino acid sites between SARS-S and human ACE2 (hACE2) [5,7,9,10] differ from those between SARS-2-S and hACE2 [11,12,13,14]. This may explain why some antibodies that are effective against SARS-S are not effective against SARS-2-S [4], especially those developed to target the ACE2 binding site of SARS-S [15]. In this article, numerous experiments on SARS-S are considered to facilitate comparisons and to highlight differences between the two.

## 2. General Features of SARS-S and SARS-2S

SARS-2-S is 1273 aa long, in contrast to 1255 aa in SARS-S. Individual protein domains in the S protein tend to fold independently and are associated with specific functions. The numbers (Figure 1A) that indicate the start/end of individual domains in SARS-S and SARS-2-S may mislead readers to think that the boundary is based on some clearly recognizable physiochemical landmarks. In fact, these numbers are for rough reference only. For example, the boundaries of RBD in SARS-S mainly result from experiments with different RNA clones containing different parts of RBD [17,18,19]. The 5′ side is delimited by the site where upstream mutations/deletions do not affect receptor binding, but downstream mutations/deletions do affect receptor binding. Similarly, the 3′ side is where upstream mutations/deletions affect receptor binding, but downstream mutations/deletions do not have an effect. Boundaries of some domains are substantiated by protein structure, for example, the boundaries of RBD [11,12,13,14,20], but some are not substantiated by protein structure.

Some inter-domain segments (Figure 1C,D) could be much more conserved than neighboring domains. For example, C822, D830, L831, and C833 in SARS-S (corresponding to C840, D848, L849, and C851 in SARS-2-S) are located between FP and IFP but are highly conserved and critically important for membrane fusion [21]. Similarly, V601 in SARS-S (corresponding to V615 in SARS-2-S) does not belong to any recognized domain (Figure 1A) but is highly conserved. Replacing it by G contributes to viral escape from neutralizing antibodies [22]. Experimental mutations at sites 1111–1130 in SARS-S, upstream of HR2 (Figure 1A), are also associated with viral escape from neutralizing antibodies [23], suggesting that mutations at those sites affect protein structure. This segment is highly conserved in SARS-2-S and related viruses, and antibodies targeting this region provide broad protection against heterogeneous viral strains [23]. In short, inter-domain segments may not be functionally less important than those recognized domains, and the sequences in these inter-domain regions are no less conserved than those within domains. More studies will reveal their functions leading to more detailed structure-function maps.

Experiments with a truncated SARS-S excluding the C-terminus indicates that it is synthesized in the endoplasmic reticulum (ER), modified in the Golgi apparatus, glycosylated, and eventually exported to the membrane [24]. Spike protein synthesis following SARS-CoV infection can cause an unfolded protein response (UPR) [25], suggesting its association with the ER. The UPR restores ER homeostasis by upregulating chaperone proteins to increase the protein-folding capacity in the ER and by reducing translation and increasing protein degradation to reduce the folding load (review in [26]). When prolonged UPR fails to restore ER homeostasis, it often triggers apoptosis. Adenovirus-mediated overexpression of S2 induces apoptosis [27] and may have implications for viral pathogenicity and secondary bacterial infection.

Coronavirus S proteins are heavily glycosylated with 21–35 N-glycosylation sites [17]. Replacing these N-glycosylation sites in SARS-S alters protein folding and expression [18]. Glycosylation events have been identified mainly in two ways. The first way has been to compare the expected molecular weight of an expressed segment of S protein containing a putative N-glycosylation site against the actual molecular weight [18]. An increase in the actual molecular weight is assumed to be due to N-glycosylation. The second way has been by high resolution mass spectrometry [28]. O-glycosylation was also found in SARS-2-S [28]. Glycosylation is not required for receptor-binding in SARS-S [18] or MHV (murine hepatitis virus) [17].

## 3. Cleavage Sites

The S protein undergoes two crucial cleavage events, with the first splitting S1 and S2 and the second splitting S2 into FP and S2′ (Figure 1A). The most pronounced difference between SARS-S and SARS-2-S is an additional furin cleavage site (site 1, Figure 2A) resulting from an insertion of 12 nt at the boundary between S1 and S2 [8,11,29]. This additional furin cleavage site is shared among all sequenced SARS-CoV-2 genomes, but absent in all their closest known relatives such as bat RaTG13 and those isolated from pangolin [29]. The seemingly sudden appearance of this additional polybasic furin cleavage site 1 has been a lasting source of conspiracy theory that SARS-CoV-2 is man-made, which is discussed later.

The furin cleavage site was predicted in February 2020 [8] and, in May 2020, its functional importance was confirmed, i.e., that the cleavage was essential for efficient viral entry into human lung cells, especially in cell-cell fusion to form syncytium to facilitate viral spread from one cell to another [2]. This exemplifies the rapidity in the progress of SARS-2-S research.

The cleavage of the S protein into S1 and S2 is an essential step in viral entry into a host cell, and needs to occur before viral fusion with the host cell membrane [6]. Different cleavage sites targeted by different proteases are often associated with drastically different virulence and host cell tropism in various RNA viruses. For example, the low-pathogenicity forms of the H1N1 influenza virus has a cleavage site by trypsin-like proteases [31] in contrast to the high-pathogenicity forms with a furin cleavage site cleaved by furin-like proteases [32]. Trypsin-like proteases typically have a narrow tissue distribution in humans. For example, trypsin-like transmembrane serine protease 11D (gene name TMPRSS11D) is expressed only in the esophagus (Figure 2C). Another member of the trypsin family, PRSS1, is expressed mainly in the pancreas [30]. In contrast, furin-like proteases are ubiquitous (Figure 2C). Thus, if a coronavirus needs to be cleaved TMPRSS11D or PRSS1, then its cellular entry is limited to the esophagus where TMPRSS11D is expressed (Figure 2C) or the pancreas where PRSS1 is expressed. However, if the virus gains a furin cleavage site, then this restriction is removed because FURIN is ubiquitous in human tissues (Figure 2C), resulting in dramatic broadening of host cell tropism. In this context, the S protein contributes to host specificity [6], and also to tissue specificity through its differential requirement of tissue-specific proteases. For this reason, viruses with different cell tropism may accumulate tissue-specific genomic signatures [33].

Because the C-terminus of the spike protein is anchored inside the viral membrane, one might expect the distal S1 to be lost after cleavage at site 1. However, the distal S1 subunit remains non-covalently bound to the S2 unit in the prefusion conformation after cleavage at site 1 [10,11,34]. In order to stabilize the prefusion conformation to facilitate vaccine design [10,35] or structural determination [11,12], the furin site is often mutated so that it is not cleaved. For example, the cleavage site RRAR was changed to GSAS in obtaining protein structure 6VSB [12], and to SGAG in obtaining protein structures 6VXX and 6VYB [11].

The cleavage site 2 (Figure 2A) is highly conserved in all sequenced SARS-CoV-2, as well as in all its close relatives including SARS-CoV. This site is likely cleaved by cathepsin L in endosome in both SARS-S [34,36,37,38] and SARS-2-S [4]. Cathepsin L requires an aromatic residue at P2 and a hydrophobic residue at P3 [39]. Cleavage site 2 has Y at P2 and A at P3 to satisfy this requirement (Figure 2A). The low pH in endosomes is optimal for cathepsin L activity. Inhibitors of cathepsin L block SARS-CoV infection [36]. 

While cleavage site 1 (Figure 2A) is known to be cleaved during SARS-CoV-2 assembly, most likely by furin in the Golgi apparatus [2,11,24,40], it is less clear how cleavage site 2 (Figure 2A) is used in SARS-2-S priming. One could hypothesize if cleavage site 1 is efficient [2], then cleavage site 2 would seem redundant and may accumulate mutations in the *SARS‑2‑S* gene without a negative impact on the fitness of the virus. However, the amino acid sites near site 2 (VASQSIIAYT|MSLGAEN, where the vertical bar indicates the scissile bond, Figure 2A) was perfectly conserved among all SARS-2-S sequenced by 8 May 2020. In contrast, each site of the 4-AA insertion (PRRA, Figure 2A) has experienced at least one amino acid replacement. Thus, in spite of the additional furin cleavage site 1, cleavage site 2 (Figure 2A) may still be functionally important for it to be so evolutionarily conserved. 

In addition to site 1 and site 2 (Figure 2A) that cleave SARS-2-S into the S1 and S2 domains, a third cleavage site also exists for cleaving S2 into FP and S2′ domains (Figure 2B,D). This site, often referred to as the S2′ site, is likely cleaved by TMPRSS2 [41,42,43,44], consistent with the finding that TMPRSS2 is needed for SARS-CoV-2 infection [3]. In particular, TMPRSS2 needs to be expressed in the target cell for it to be infected [41]. Because TMPRSS2 is active mainly in the membrane or extracellular space, the third cleavage site is not cleaved during SARS-CoV assembly [24,41]. This site can also be cleaved by trypsin. Exogenous trypsin can enhance membrane fusion and SARS-CoV infection [45,46]. Trypsin cleaves SARS-S at R797 (Figure 2D), consistent with the finding that an R797N mutation abolishes this trypsin-induced membrane fusion [34]. 

The temporal sequence of cleavage events is not clear, although the following order is likely: For SARS-2-S, furin cleaves at cleavage site 1 during viral assembly [2]. Then, the third cleavage site is cleaved by TMPRSS2 to yield FP and S2′ (Figure 2D) to trigger membrane fusion, syncytium formation, and viral entry into a target cell [3,11,34]. For SARS-S, cleavage site 1 does not seem to be used efficiently. The transmembrane TMPRSS2, if expressed, cleaves the third cleavage site to yield FP and S2′ and to trigger cell fusion and viral entry [3]. This may be termed the membrane-TMPRSS2 pathway of viral entry. If SARS-S is not cleaved by TMPRSS2 into FP and S2′, then the virus can enter the cell through endocytosis with cleavage site 2 cleaved by cathepsin L. This is the endosome-cathepsin pathway of viral entry [41,46].

## 4. The S1 Domain

### 4.1. The Signal Peptide

The spike protein requires a signal peptide (SP) to guide its transportation to its membrane destination. The SP consists of the first 13 amino acids with helix-forming high-hydrophobicity residues (Figure 1F), as is typical of almost all signal peptides. The only other SARS-S segment of high hydrophobicity is the transmembrane domain (TM, Figure 1F). These two hydrophobic regions at the two extremes of S are shared among diverse betacoronavirus lineages. The SP from different coronaviruses are only weakly homologous at the nucleotide or amino acid level (Figure 2B), but they share helix structure and high hydrophobicity in common.

### 4.2. The N-Terminal Domain (NTD)

The function of *N*-terminal and C-terminal domains of S1 differs among different betacoronavirus lineages. The receptor for S protein in MHV is carcinoembryonic antigen cell-adhesion molecules (CEACAMs), and the receptor-binding domain is near the *N*-terminal [47,48]. As receptor binding is clearly a vital function for any coronavirus, MHV’s NTD is conserved with no indels in aligned MHV S protein sequences, whereas its C-terminal domain homologous to RBD in SARS-S and SASRS-2-S is littered with many indels. For most betacoronaviruses, RBD is near the C-terminus of S1 (Figure 1A), and this RBD domain tends to be more conserved at the nucleotide and amino acid level, and also in the sliding-window hydrophobicity plot (Figure 3A) than in the NTD.

### 4.3. The Receptor-Binding Domain (RBD)

The RBD domain has a core subdomain and a receptor-binding motif that directly interact with the host ACE2 [4,5,50]. It has been used extensively as a drug target for anti-SARS-CoV drug and vaccine development [51,52,53,54,55,56,57]. A good vaccine should be safe but highly immunogenic and should not become obsolete as soon as there are viral mutations. RBD-based vaccines have been found to be highly immunogenic [58,59], even when they are expressed in yeast [60], which suggests that they fold independently of other parts of the spike protein and that the folding is robust in different folding environments. However, it is more difficult to establish the safety and long-term effect of the vaccines. 

In spite of much effort to develop drugs and vaccines based on RBD, there is an inherent problem with this approach because RBD is highly variable at the sequence level [17]. The sequence variability in S1 relative to S2 is also highly visible in a sliding-window isoelectric point (pI) plot (Figure 3B). Because of high variability in S1 among different viral species, RBD-based antibodies or vaccines developed against SARS-CoV [54,55,56] typically do not offer heterologous protection against other coronaviruses such as MERS-CoV [61]. In fact, some antibodies against SARS-CoV strains in the first viral outbreak were no longer effective against SARS-CoV in the second outbreak [62], cautioning against drug development targeting variable domains. In contrast, human monoclonal antibodies against the more conserved S2 are expected to be more broadly neutralizing, which is true as demonstrated with antibodies against highly conserved HR1 and HR2 domains of SARS-S [63]. Thus, given that a virus can escape neutralizing antibodies by just a single amino acid replacement [22,64], one should develop anti-viral drugs or vaccines by targeting only highly conserved regions.

## 5. The S2 Domain

While the S1 domain mainly functions in receptor binding, the S2 domain functions mainly in membrane fusion. They represent two distinct steps in SARS-CoV infection [20,36] and SARS-CoV-2 infection [2,3,8]. This S2 function of membrane fusion was inferred early because many antibodies targeting S2 of coronavirus S proteins were almost always associated with disrupted membrane fusion [17]. Vaccine targeting segments 884–891 and 1116–1123 in S2 were highly effective in inducing humoral and cell-mediated immune responses [65]. These segments belong to the central helix between HR1 and HR2. However, some antibodies targeting S2 have been shown to be cytotoxic [66].

Membrane fusion requires two anchors, one at the virion side and the other at the host cell side [67,68]. In the case of SARS-S and SARS-2-S (Figure 1A), the C-terminus is anchored inside the virion, and the FP domain of S2 (or IFP domain of S2′ when FP is cleaved off) penetrates the target cell membrane to install the anchor inside the target cell [67,69]. 

Membrane fusion appears to have two distinct types associated with different pathways of cell entry. The first type involves a virus in a non-cellular environment (e.g., in the airway of human respiratory system) finding its way inside an epithelial cell, and the second type involves a virus in an infected cell finding its way to a neighboring cell. The first type would require fusion of the viral membrane and the target cell membrane, and the second type would be facilitated by the formation of syncytium through the cell-cell fusion [2,34]. 

### 5.1. Fusion Peptides

Many viral fusion proteins exist [67,70]. All known viral fusion peptides form trimers [67], but they often exhibit little sequence homology among different viral species, suggesting evolutionary convergence in trimer formation. The S protein needs a trigger to induce conformational change for membrane fusion [67], and the trigger is typically a cleavage event that occurs either at the cell surface at neutral pH or within an endosome at a reduced pH. These correspond to the two viral entry pathways in SARS-S and SARS-2-S, i.e., the membrane-TMPRSS2 pathway and the endosome-cathepsin L pathway [41,42,43,44,46].

The FP (or IFP when FP is cleaved off) in SARS-S and SARS-2-S (Figure 1A) serves to penetrate the target cell membrane and install an anchor inside. The TM and CT domains (Figure 1A) form an anchor inside the virion. The S2 (or S2′) between the two anchors will undergo conformational change to bring the two membranes together for fusion. The conformational change needs to be triggered by a signal that should reliably indicate the proximity between a virus and a target cell or between an infected cell and a target cell. The triggering signal most likely is TMPRSS2 expressed on the surface of a target cell [41,42,43,44,46]. Thus, the cleavage of S2 at 797R|S798 in SARS-S (where|indicates the scissile bond) or 815R|S816 in SARS-2-S (Figure 2D) by TMPRSS2, exposing IFP at the *N*-terminal of S2′, appears to be a reliable signal to the virus that a good target cell is within reach. This is consistent with the finding that a target cell needs to express TMPRSS2 to be infected, but altering expression of TMPRSS2 in the infected cell does not affect the efficiency of infection [41]. If no TMPRSS2 cleaves S2, then viral entry may go through the endosome-cathepsin L pathway in which endocytosis occurs resulting in S2 cleaved into FP and S2′ in endosome to trigger membrane fusion. Further research is needed to substantiate and validate the details.

### 5.2. The Heptad-Repeat Domains: HR-1 and HR-2

Heptad repeats (HR, Figure 1E) are characterized by repeated 7mers represented as (abcdefg)_n_ with amino acids at positions a and d being hydrophobic. In leucine zipper transcription factors such as GCN4 in yeast [71] and XBP1 in humans [72], the d positions are occupied exclusively by leucine [73]. HRs are relatively poor in glycine (which would permit too much bending flexibility). They form helices, contain no helix-breaking prolines and no clustered charged residues, and are typically located next to hydrophobic fusion peptides in RNA viruses [74]. Hydrophobic residues, at positions a and d, are on the same side of the helix (Figure 1E) and form a hydrophobic interface with other helices. Because SARS-S and SARS-2S are homotrimers, there are three HR1 and three HR2 forming a six-helix bundle [6,68,75]. The six-helix bundle is also observed in SARS-2-S [76]. It has been inferred that helices formed from HRs are perpendicular to the viral membrane [74], which has been substantiated in both SARS-S and MERS-S [6].

Given that a viral HR typically follows an *N*-terminal hydrophobic region in diverse viral lineages [74], one may infer that such a configuration is favored by natural selection to serve the function of membrane fusion. In this context, the configuration of (FP + IFP + HR1) may not be as favorable as that of (IFP + HR1), the latter resulting from cleavage at the third cleavage site (Figure 2E) to split S2 to FP and S2′. This may explain why the cleavage at this site dramatically enhances membrane fusion and viral entry [3,37,41].

HR1 and HR2 are strongly conserved among SARS-S, SARS-2-S and their relatives (Figure 1E). The isoelectric point along a sliding-window is essentially identical among the six viral strains in regions from HR1 to CT (Figure 3B), in contrast to that for S1 where much scatter is observed. Structural comparisons have revealed conservation of HR1 in multiple coronaviruses [77]. Partly for this reason, antibodies have been developed that target these regions [23,63,78]. Such antibodies typically provide broad protection against multiple viruses [23,63], because sequences in this region are highly conserved. A previously developed pan-coronavirus fusion inhibitor (EK1) against HR1 in SARS-S to inhibit membrane fusion was also found to inhibit membrane fusion during infection by SARS-CoV-2 and MERS-CoV [76]. Thus, drug repurposing of anti-SARS-S drugs for fighting against SARS-CoV-2 should focus on drugs or vaccines targeting highly conserved regions of SARS-S.

Individual helix-forming segments in HR1 and HR2 can bind to each other, which creates an opportunity to use such HR1 and HR2 segments as drugs to disrupt membrane fusion [68,75]. HR2 peptides have been used to inhibit infection by MHV, but this inhibition is less effective against SARS-CoV [68].

The segment between HR1 and HR2 (Figure 3) is the central helix. There is a transitional bend between HR1 and the central helix which, when fixed with two consecutive proline residues, prevents structural transitions from prefusion to postfusion, and consequently contributes to the stabilization of the spike protein at the prefusion state which is important for vaccine development [10,35,79]. Spike proteins with these two proline replacements are known as S-2P. This is discussed further in the section on vaccine development.

### 5.3. The Transmembrane Domain and Cytoplasmic Tail Domain

The transmembrane (TM) domain of the S protein (Figure 1A) is known to be highly conserved in SARS-CoV-2 and its close relatives [69]. This conservation is also reflected in the hydrophobicity profile and pI profile among SARS-CoV-2 and its close relatives (Figure 3). The TM domain consists of the following three parts [69,80]: a juxtamembrane aromatic part, a central hydrophobic part, and a cysteine-rich part (Figure 4). It is followed by a highly hydrophilic cytoplasmic tail (CT) which anchors the spike inside the viral membrane.

The tryptophan residues in the aromatic part are strongly conserved among SARS-CoV-2 and related coronaviruses, suggesting their functional importance. Replacing them even by another aromatic residue such as phenylalanine will severely impact the efficiency of viral infection [80]. However, this finding was not supported in another study [81] in which replacing tryptophan by phenylalanine was tolerated. 

The central hydrophobic part forms a helix. Because S proteins form a homotrimer, there are three transmembrane helices interacting with each other. The TM and the C-terminus contribute to the stabilization of the trimeric structure [19,24,69] which is important for membrane fusion. Destabilization of the trimeric structure is associated with reduced fusogenicity and infectivity [69]. Replacing hydrophobic residues in the central part by hydrophilic ones such as lysine decreases the efficiency of an infection [80]. Cysteine residues immediately proximal to the membrane (near the central hydrophobic part in Figure 4) are palmitoylated; replacing them by other amino acids (e.g., alanine) inhibits membrane fusion [82]. In contrast, replacing cysteine residues in the last half of the cysteine-rich part or even deleting them does not inhibit membrane fusion [82,83]. 

During the cell-to-cell infection stage, the membrane-proximal cysteine-rich part, and the cytoplasmic tail anchor the C-terminus of S inside the infected cell, and the *N*-terminal of S2 (or S2′) penetrates the membrane of a target cell and anchor the *N*-terminus inside, which is typical of viral fusion proteins [67]. The conformational changes of S2 (or S2′), including the tripartite TM, help to bring the membranes of infected and target cells close together to facilitate cell-cell membrane fusion and viral entry [2,34,80,84]. The anchor provided by the cysteine-rich part and CT is enhanced by the membrane-actin linker ezrin [84] which, upon phosphorylation, links specific transmembrane proteins such as S homotrimer to actin to reinforce the anchor inside the cell.

## 6. The Spike Protein in Vaccine Development

Almost all vaccine candidates against SARS-CoV-2 are based on the spike protein, including the FDA-approved Pfizer/BioNTech and Moderna vaccines that use mRNA encoding a modified spike protein stabilized in its prefusion conformation. It is important for the immune system to respond to the virus at the prefusion stage, because it would probably be too late for the immune system to intervene at the postfusion stage when the virus is gaining entry into an uninfected cell. Therefore, the rationale of vaccine development is to produce a spike protein stabilized in the prefusion conformation as a target to train the immune system to act against it.

Two structural studies on spike proteins, one on Betacoronavirus HKU1 [10] and the other on MERS-CoV [79], have demonstrated that replacing two consecutive amino acids by proline near the transition from HR1 to the central helix (Figure 3) would strongly contribute to the stabilization of the resulting spike protein at the prefusion conformation. These amino acid sites correspond to sites 986 and 987 in SARS-2-S (Figure 5), located at the transitional bend between HR1 and the central helix (Figure 3). Amino acids at two sites are not conserved, being NL in CoV-HKU1, VL in MERS-CoV, and KV in SARS [79], suggesting that they are probably not functionally important. However, the two amino acid replacements (K986P, V987P), shown in Figure 5, stabilize the resulting spike protein in the prefusion state and contribute to vaccine efficiency. The mutant SARS-2-S spike protein with these proline replacements is referred to as S-2P [85,86], which is encoded in the mRNA vaccine from both Pfizer/BioNTech (BNT162b2) and Moderna (mRNA-1273). A new spike protein variant (HexaPro) that includes four additional amino acid replacements by proline (F817P, A892P, A899P, and A942P) is even more stable and expressed more than the original S-2P [35].

## 7. Structural Insights into the Emergence of New Viral Variants

Here, one example is described to illustrate how structural biology can shed light on the emergence of new viral variants. In an experiment that used neutralizing monoclonal antibodies to select neutralization-escaping SARS-CoV variants [22], one of the four variants was V601G within SARS-S at 594VAVLYQD**V**NCTDV606 where V601 was highlighted in bold. The identification of this infection-enhancing V601G variant is puzzling because one does not expect that such a V→G replacement would have much phenotypic effect on the S protein. First, site 601 is not involved in receptor binding. Second, both V and G are small and nonpolar. Therefore the replacement is conservative and should not cause a significant structural perturbation of the S protein. Does a replacement of a small nonpolar V by a smaller nonpolar G really matter? One cannot answer the question without structural evidence. It can only be inferred that site 601 is functionally important, and that the smallest amino acid at site 601 (or its vicinity) is beneficial to SARS-CoV.

A V601G mutation requires a transversion (i.e., from codon GUN to GGN). Because of proofreading in coronavirus genome replication [87,88,89], transversional mutations are much rarer than transitions. For this reason, V→G at site 601 is expected to occur much more frequently than D→G at site 600, because the latter requires a transition (from codon GAY to GGY) instead of a transversion. Therefore, a small G can be gained by a D600G mutation instead of a V601G mutation. The segment of 594VAVLYQD**V**NCTDV606 in SARS-S corresponds to 608VAVLYQDVNCTEV620 in SARS-2-S, therefore, a D600G mutation in SARS-S is equivalent to D614G in SARS-2-S. In this context, it is not surprising that a D614G variant of SARS-CoV-2 quickly increased in frequency [90], indicating a strong selective advantage.

Now, there are two alternative hypotheses concerning the selective advantage of the D614G mutation as follows: (1) the benefit is due to G being the smallest amino acid, or (2) the benefit is due to the loss of a negative charge altering electrostatic interactions. The second hypothesis may be dismissed on the following empirical grounds: Codons encoding D (GAY) could also mutate to AAY encoding N through a single transition. Such a mutation would lose the negative charge carried by D. If it is the loss of a negative charge that is beneficial, we would expect AAY and GGY to be roughly equally represented at this site. However, AAY is entirely missing in sequenced SARS-2-S, which goes against the second hypothesis. Unfortunately, exclusion of the second hypothesis neither implies confirmation of the first (because there are other alternatives), nor helps us understand why the D614G mutation enhances viral fitness. Only through structural studies [91] can we hope to gain a mechanistic understanding of the effect of the D614G mutation on the S protein.

## 8. The Spike Protein and the Conspiracy Theory

As previously mentioned, the additional polybasic furin cleavage site 1 (Figure 2A) has been a lasting source of conspiracy theory that SARS-CoV-2 is man-made. Advocates of the conspiracy theory assume that scientists have ignored or refused to address their legit concerns. In this review, two points are made. First, the evidence for a natural origin of SARS-CoV-2 is accumulating, albeit at a rate slower than desired. Second, the reasons behind the conspiracy theory have been seriously considered by scientists and have been deemed to be not strong reasons.

There are three main reasons for the conspiracy theory, all involving the polybasic furin cleavage site (Figure 2A). First, the furin cleavage site has not been observed in any close relatives of SARS-CoV-2 in nature. A somewhat similar furin cleavage site was present at a roughly homologous site in S protein of the murine hepatitis virus [45] and in a few alphacoronaviruses [2,8,29]. However, it is not clear how SARS-CoV-2 could gain it from these remote relatives. While recombination might be a possibility, there is hardly any sequence homology between SARS-2-S and its homologues in the murine hepatitis virus or alphacoronaviruses at sequences flanking the cleavage site, therefore, a recombination origin of the cleavage site is tenuous at present. An insertion at the same site was found in a bat-derived coronavirus [92], but the inserted sequence was different and could not function as a furin cleavage site. A novel bat-derived coronavirus (RmYN02) was reported to have an insertion bearing a weak semblance to the polybasic furin cleavage site in Figure 2A [92], suggesting the possibility of a natural origin of the polybasic furin cleavage site. However, the sequence homology between RmYN02 and SARS-2-S is low, and it is not clear if the insertion in RmYN02 is real or an artefact of alignment. Therefore, if one cannot offer a plausible hypothesis of natural origin of the polybasic site, it is easy to fall back on the hypothesis of artificial origin. This reminds us of the period of time before Darwin, i.e., when the origin of species cannot be fully explained, it is easy to fall back to the theory of a creator.

The second reason for the conspiracy theory is associated with the feasibility of creating such a polybasic site and a need to create such a site for testing certain biological hypotheses. Some background information arising from SARS-S is needed to understand this reason. The roughly homologous RNA segment in SARS-S is a weak cleavage site, likely cleaved by transmembrane serine protease TMPRSS2 [93]. R667 in SARS-S (immediately upstream of the site 1 cleavage in Figure 2A) is required for cleavage by TMPRSS2 [93]. The site can also be cleaved by trypsin, and processing of SARS-S by trypsin enhances viral infectivity [34,45,94]. Because trypsin and trypsin-like proteases are strongly tissue restricted (Figure 2C), the site is typically not cleaved in SARS-S [24]. It is natural for one to hypothesize that adding a furin cleavage site would allow the site to be efficiently cleaved in nearly all tissues, potentially enhancing SARS-CoV infection and broadening its cell tropism. Indeed, introducing a furin cleavage site at the S1 and S2 boundary of SARS-S has increased cell-cell fusion (syncytium formation) and viral infectivity [34]. This result suggests that the additional polybasic furin cleavage site may have contributed significantly to the efficiency of SARS-CoV-2 in infecting human. Host cells, in response to viral infection, may reduce furin activities [8]. 

In short, given the seemingly sudden appearance of the additional furin cleavage site that cannot be readily explained by a hypothesis of natural origin, and the fact that virologists have already experimented with adding a furin cleavage site at this specific location and learned the consequence of enhanced viral infectivity and cell-cell fusion, the claim that the polybasic furin cleavage site in SARS-2-S has been experimentally inserted is not too far-fetched. However, the global collaboration among scientists, in general, and virologists, in particular, has created scientific communities that are far more closely knit than before. While it is possible to create a viral pathogen, it is extremely unlikely for a laboratory to create SARS-CoV-2 without being noticed. 

The third reason is that the 12 nt insertion encoding the polybasic furin cleavage site carries two CpG dinucleotides. Such CpG dinucleotides are very rare in SARS-CoV-2 [95], and particularly rare in SARS-2-S. Why would such CpG rarity contribute to the conspiracy theory? Mammalian zinc finger antiviral protein (ZAP, gene name ZC3HAC1) targets CpG dinucleotides in viral RNA to mediate RNA degradation and inhibit viral replication [96]. The ZAP-mediated RNA degradation is cumulative [96], as shown by the following experiment. When CpG dinucleotides were experimentally added to individual viral segment 1 or 2, the inhibitory effect of ZAP was weak. However, when the same CpG dinucleotides were added to both segments 1 and 2, the ZAP inhibition effect was strong [96]. This implies that only mRNA sequences of sufficient length would be targeted by ZAP (i.e., *S*, *1ab,* and *1a* mRNAs in SARS-CoV and SARS-CoV-2). SARS-CoV-2 and its closest relatives from bat (RaTG13) and pangolin exhibit the strongest genomic CpG deficiency among all betacoronaviruses [95], presumably to evade ZAP-mediated host defense. The *S* gene is particularly CpG-deficient as measured by two indices, I_CpG_ [95,97] and ln (N_CG_/N_GC_) (Table 1), where N_CG_ and N_GC_ are the numbers of CpG and GpC dinucleotides in the *S* gene. I_CpG_ < 1, or ln (N_CG_/N_GC_) < 0, means CpG deficiency.

Because of this ZAP-mediated selection against CpG, SARS-CoV-2 and its close relatives encode most of arginine residues by the two AGR codons, instead of the four CGN codons. The *S* gene encodes 42 arginine residues, with only 12 (28.57%) encoded by the four CGN codons in contrast to 30 encoded by the two AGR codons. The two arginine residues in the polybasic furin cleavage site are encoded by the rare CGN codons, which seems unnatural in this context. However, the probability of randomly picking up two arginine codons that happen to be both CGN codons is not extremely low (i.e. =0.2857^2^ = 0.0816).

One way to dispel the conspiracy theory is to find a set of viral lineages in wildlife that would allow reconstruction of a plausible evolutionary path leading to the origin of the polybasic furin cleavage site. The “missing link” that would satisfy conspiracy theorists is still to be found. However, there is no guarantee that it will be found because nature is not obliged to preserve all what she has created.

## 9. Conclusions

In summary, although much is known about the S protein in coronaviruses, the temporal and spatial changes of S during synthesis, glycosylation, cleavage, membrane fusion, and viral entry remain poorly defined. It is also important to keep in mind that the S-mediated cell entry is only one step in the viral infection cycle and naturally cannot explain all differences in virulence among betacoronaviruses. For example, MERS viruses found in Africa exhibit reduced replicative capability and are typically not pathogenic relative to the prototypic and highly pathogenic Arabian MERS-CoV strain. However, the two are not different in their efficiency in gaining host cell entry [98], pointing to differences in other parts of the viruses that may contribute to their differences in pathogenicity. 

## Figures and Tables

**Figure 1 viruses-13-00109-f001:**
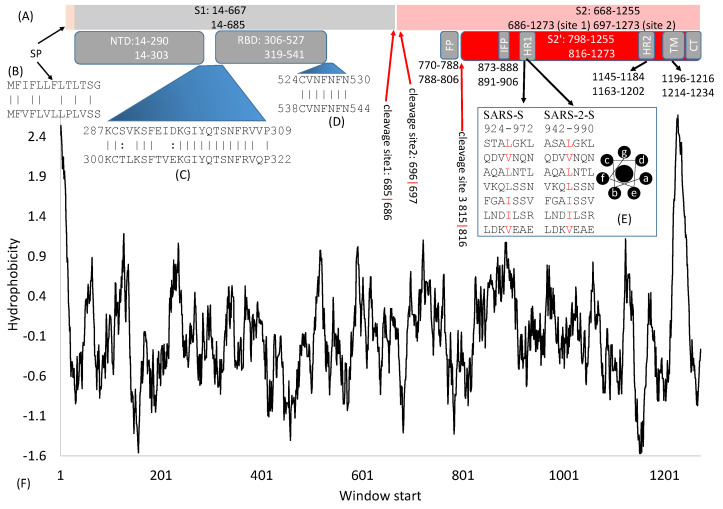
Domain structure of SARS-S and SARS-2-S. (**A**) Key domains in SARS-S and SARS-2-S. SP, signal peptide; NTD, *N*-terminal domain; RBD, receptor-binding domain; FP, fusion peptide; IFP, internal fusion peptide; HR, heptad repeats; TM, transmembrane domain; CT, cytoplasmic tail. The top and bottom numbers in each domain pertain to SARS-S and SARS-2-S, respectively. The red arrows indicate cleavage sites, and their numbers pertain to SARS-2-S; (**B**) Alignment of SP between SARS-S (top) and SARS-2-S (bottom); (**C**,**D**) Alignment of two inter-domain segments; (**E**) HR1 in SARS-S and SARS-2-S, together with the top view of a helix showing hydrophobic positions a and d on the same side; (**F**) Hydrophobicity plot generated from DAMBE [16].

**Figure 2 viruses-13-00109-f002:**
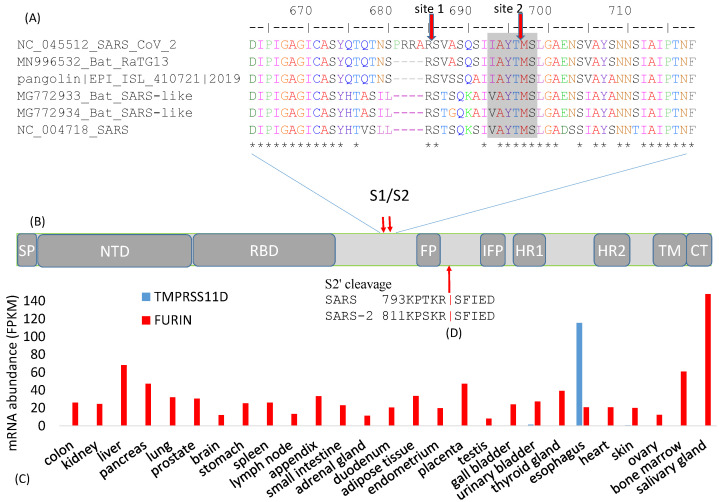
Cleavage sites at the S1/S2 boundary. (**A**) An insertion of 12 nt in SARS-CoV-2 results in a new polybasic furin cleavage site, resulting in two cleavage sites indicated by the red downward arrows. “*” indicates sites that are identical among the six viral strains. Numbers follow (**B**) Schematic domain structure of S protein, with the same abbreviation as in Figure 1A; (**C**) Tissue-specific mRNA distribution of human trypsin-like protease TMPRESS11D and FURIN, derived from [30]; (**D**) Cleavage site for splitting S2 into FP and S2′ domains.

**Figure 3 viruses-13-00109-f003:**
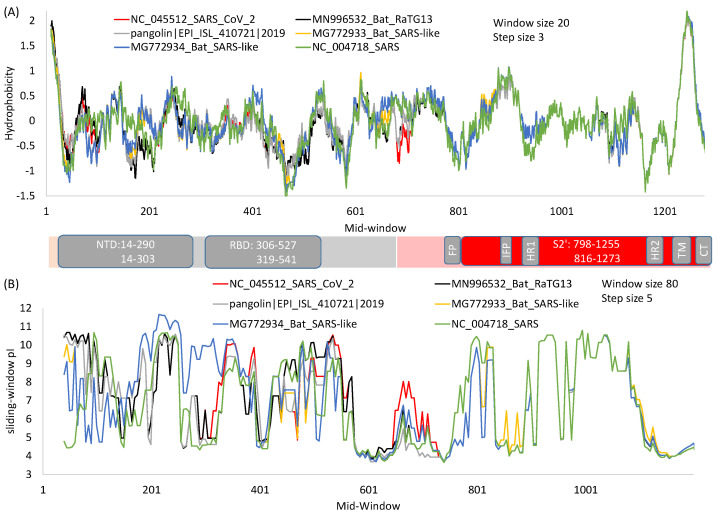
Hydrophobicity (**A**) and protein isoelectric point (**B**) plots of spike protein from SARS-CoV-2 and its close relatives over sliding windows. For window-specific calculation of isoelectric point (pI), the *N*-terminus amino group is added to the first window and the *C*-terminus carboxyl added to the last window. Generated from DAMBE [49].

**Figure 4 viruses-13-00109-f004:**
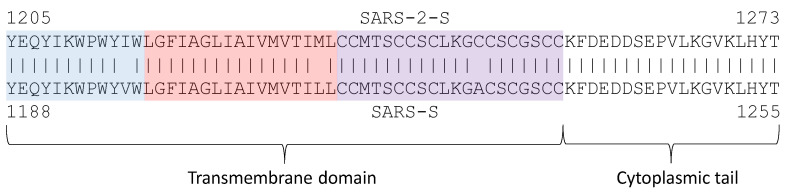
Transmembrane (TM) domain with its tripartite structure (juxtamembrane aromatic part in blue, central hydrophobic part in pink, and cysteine-rich part in purple) and the cytoplasmic tail that anchors inside the viral membrane.

**Figure 5 viruses-13-00109-f005:**
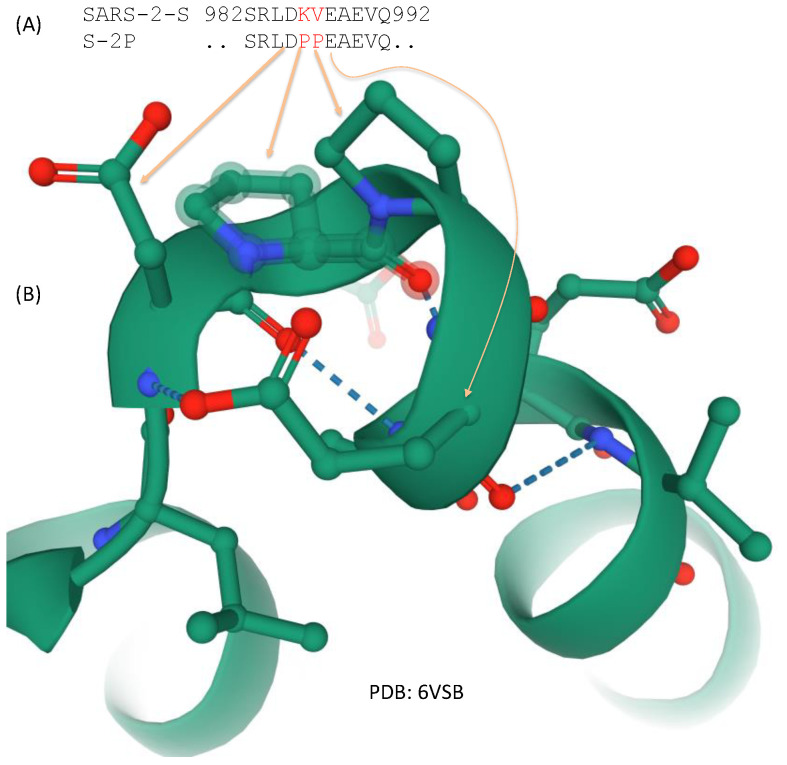
Two amino acid replacements that stabilize the spike protein at the prefusion state. (**A**) Amino acids KY in the native state of SARS-2-S is replaced by PP spike variant S-2P used in the FDA-approved Pfizer/BioNTech and Moderna vaccine; (**B**) Partial structure from 6VSB showing the two proline residues stabilizing the structural bend.

**Table 1 viruses-13-00109-t001:** Genomic CpG deficiency in the coding sequence encoding the spike proteins, measured by two indices: I_CpG_ = (P_C_*P_G_/P_CG_) and ln (N_CG_/N_GC_). The expectation of no CpG deficiency is 1 for I_CpG_ and 0 for ln (N_CG_/N_GC_).

Sequence Name	Length	I_CpG_	N_CG_	N_GC_	Ln (N_CG_/N_GC_)
NC_045512_SARS-CoV-2	3819	0.2179	29	137	−1.5527
MN996532_Bat_RaTG13	3807	0.2753	37	140	−1.3307
pangolin/EPI_ISL_410721/2019	3795	0.2857	38	139	−1.2969
MG772933_Bat_SARS-like	3738	0.3618	50	148	−1.0852
MG772934_Bat_SARS-like	3735	0.3697	50	142	−1.0438
NC_004718_SARS	3765	0.3673	52	174	−1.2078

## Data Availability

Not applicable.

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
