# Peer review of "Domains and Functions of Spike Protein in SARS-Cov-2 in the Context of Vaccine Design"

_viruses, 2021, doi:10.3390/v13010109_

Round 1

Reviewer 1 Report

The MS is interesting and provides insights into the structure of SARS-CoV and SARS-CoV-2 S protein structure and function and how they relate to vaccine design. I would recommend checking the MS for spelling and grammatical errors as well as typos.  

Author Response

There are indeed spelling and grammatical errors as well as typos. Thanks. I have corrected them. 

Reviewer 2 Report

This manuscript is a very carefully written review about the SARS-CoV-2 spike protein and his interaction with his receptor the ACE-2 protein. The cleaving  mechanism of the S protein in S1 and S2 is described and the functions of the cleaved subunits are explained and discussed in view of the knowledge from the literature how other viruses achieve the functions of viral cell entry and cell to cell transmission. The findings of such results described in the literature are discussed and explained and focused  on the SARS-CoV-2 situation.  The findings described are discussed in view of their importance for the SARS-CoV-2 spike vaccine development.

Very recently ( see Science from December 23th , 2020)  SARS-CoV-2 variants have been isolated from infected human beeings showing several mutations in the Spike protein. Such variants having obviously a better transmission effectivity from humans to humans, have been found in UK as well as in South Afrika.

I would like to recommend that Professor Xia should ad his opinion in an additional paragraph  about these new thrilling findings in his  manuscript.

Author Response

Reviewer 2:

Very recently ( see Science from December 23th , 2020)  SARS-CoV-2 variants have been isolated from infected human beings showing several mutations in the Spike protein. Such variants having obviously a better transmission effectivity from humans to humans, have been found in UK as well as in South Afrika.

I would like to recommend that Professor Xia should ad his opinion in an additional paragraph  about these new thrilling findings in his  manuscript.

Thanks for the suggestion. I have added one short section (Section 7) to highlight how structural studies may shed light on the D614G variant of SARS-CoV-2 that has increased rapidly in frequency. However, a thorough review of D614G and its descendant B.1.1.7 strain would require a full-length paper.